# On the Predictability of 30-Day Global Mesoscale Simulations of African Easterly Waves during Summer 2006: A View with the Generalized Lorenz Model

**Bo-Wen Shen** 

Department of Mathematics and Statistics, San Diego State University, 5500 Campanile Drive, San Diego, CA 92182, USA; bshen@mail.sdsu.edu

**Abstract:** Recent advances in computational and global modeling technology have provided the potential to improve weather predictions at extended-range scales. In earlier studies by the author and his coauthors, realistic 30-day simulations of multiple African easterly waves (AEWs) and an averaged African easterly jet (AEJ) were obtained. The formation of hurricane Helene (2006) was also realistically simulated from Day 22 to Day 30. In this study, such extended predictability was further analyzed based on recent understandings of chaos and instability within Lorenz models and the generalized Lorenz model. The analysis suggested that a statement of the theoretical predictability of two weeks is not universal. New insight into chaotic and non-chaotic processes revealed by the generalized Lorenz model (GLM) indicated the potential for extending prediction lead times. Two major features within the GLM included: (1) three types of attractors (that also appeared in the original Lorenz model) and (2) two kinds of attractor coexistence. The features suggest a refined view on the nature of weather, as follows: The entirety of weather is a superset that consists of chaotic and non-chaotic processes. Better predictability can be obtained for stable, steady-state solutions and nonlinear periodic solutions that occur at small and large Rayleigh parameters, respectively. By comparison, chaotic solutions appear only at moderate Rayleigh parameters. Errors associated with dissipative small-scale processes do not necessarily contaminate the simulations of large scale processes. Based on the nonlinear periodic solutions (also known as limit cycle solutions), here, we propose a hypothetical mechanism for the recurrence (or periodicity) of successive AEWs. The insensitivity of limit cycles to initial conditions implies that AEW simulations with strong heating and balanced nonlinearity could be more predictable. Based on the hypothetical mechanism, the possibility of extending prediction lead times at extended range scales is discussed. Future work will include refining the model to better examine the validity of the mechanism to explain the recurrence of multiple AEWs.

**Keywords:** African easterly wave; attractor coexistence; chaos; hurricane; limit cycle; Lorenz model; predictability; recurrence; extended range weather prediction

---

## 1. Introduction

Due to the pioneering studies of Lorenz [1–3], the finite predictability of weather is well accepted. Subsequent studies have focused on how to estimate the limit of predictability and reveal the fundamental mechanisms responsible for limited predictability. In those studies, a theoretical predictability limit of two weeks was suggested ([4] and references therein; [5]). As a result, some researchers have interpreted this limit as an upper bound for the intrinsic predictability of all weather systems at various scales and, thus, have determined that the practical predictability of all dynamic

models cannot be longer than two weeks. Based on recent advances in supercomputing and global modeling technology, as discussed below, promising extended-range (15–30 days) simulations have been reported. For example, as shown in Figure 1a,b, Shen et al. [6] discussed realistic simulations of tropical cyclone (TC) formation, intensification, and movement in 30-day simulations using a global mesoscale model. The simulated TC resembled the real Hurricane Helene (2006) in terms of its movement and intensification. One may wonder how such a surprising result could be obtained while the inherent limits for long-range forecasting that developed within the scientific literature remain (e.g., [4,7]). To address this question, the predictability problem that has been outstanding for decades and only partially resolved must be revisited [8,9]. Since this is not an easy task, we attempt to address this problem based on recent global mesoscale modeling studies [6,10–15], a 10-year analysis of global reanalysis data [16], and generalized Lorenz modeling studies [17–23].

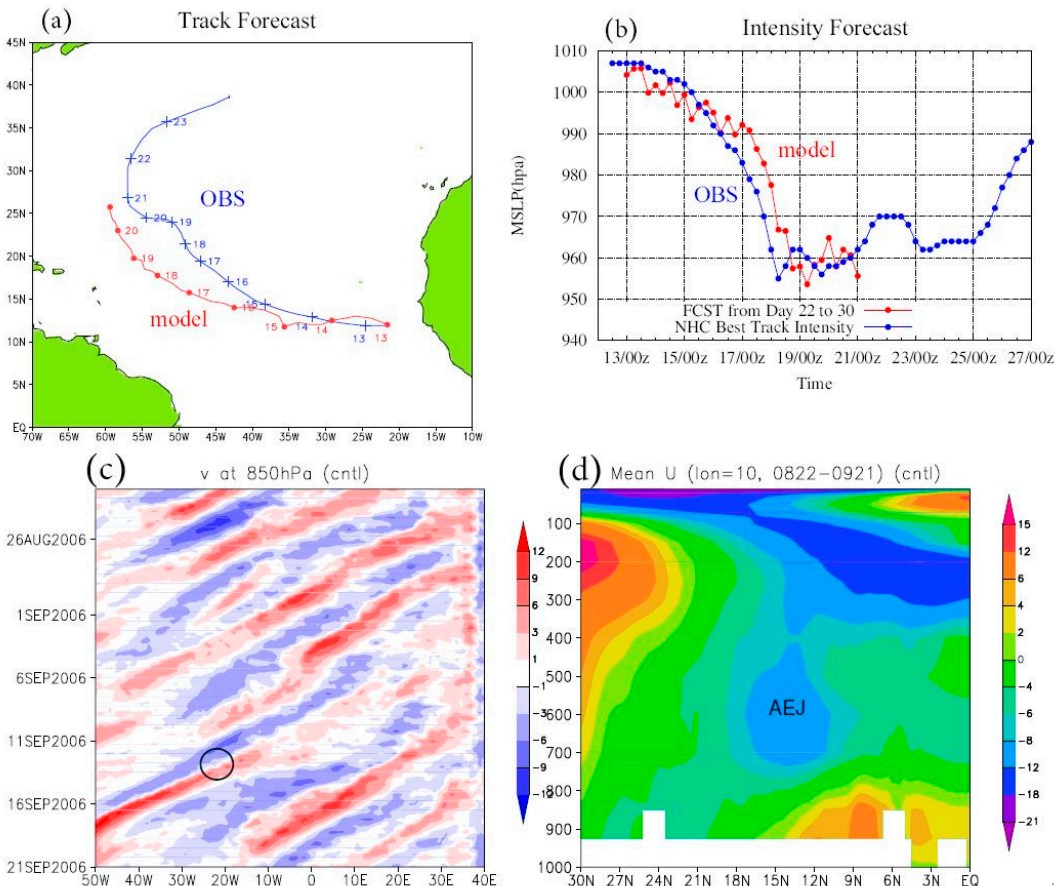

**Figure 1.** (**a**) Track and (**b**) intensity forecasts for Hurricane Helene (2006) from Day 22 to 30 in the control run initialized at 0000 UTC 22 August 2006. The red and blue lines indicate model predictions and best tracks, respectively. (**c**) Time-longitude diagrams of meridional winds averaged over latitudes 5–20° N. A black circle roughly indicated the time and longitude for the formation of Helene. (**d**) Height-latitude cross sections of time-averaged zonal winds along longitude 10° E. (courtesy of [6]).

While short-term hurricane predictions have been extended to produce promising results at 15–30 days scales, long term climate TC simulations have been shown to provide improved temporal and spatial locality for simulated TCs. For example, recent studies using numerical models and prescribed sea surface temperature have demonstrated the potential of simulating hurricane climate (such as hurricane frequency at a seasonal scale) (e.g., [24] and references therein). All of the above results are encouraging in regards to improving the accuracy of predicting hurricane formation (e.g., at a sub-seasonal scale) in long-term climate simulations. In this study, we revisit the previously published predictability problem for 30-day simulations by discussing the role of large-scale flows,

such as African easterly waves (AEWs), and small scale processes in determining the formation of hurricanes at extended-range scales.

As discussed in [25], intrinsic predictability is dependent only on the flow itself, and practical predictability is limited by imperfect initial conditions or dependence on (mathematical) formulas (e.g., [17]). Based on multi-year data analyses (e.g., [16,26]) and global mesoscale modeling simulations [6,10–14], recent studies have been conducted to understand to what extent high intrinsic predictability may exist and how the corresponding practical predictability can realistically be obtained. A conceptual model for discussing the role of hierarchical multiscale processes in the predictability of TCs and large scale waves was proposed, including: (1) downscaling processes associated with modulation due to large-scale flows such as African easterly waves (AEWs; [27]) or Madden–Julian oscillations [28] and (2) upscaling processes associated with feedbacks from small-scale processes such as convection or precipitation. Here, downscaling (or upscaling) processes indicate the transfer of energy from a large (or small) scale system to a small (or large) scale system. The conceptual model suggests the possibility of extending the lead time of TC genesis prediction by realistically simulating the evolution of large-scale processes and their modulation on TC activities, as well as feedbacks by small-scale resolved and parameterized processes.

By conducting and analyzing global mesoscale simulations for selected cases, as previously discussed, the potential impact of downscaling processes on TC simulations was discussed by revealing the relationship between: (i) TC Nargis (2008) and an equatorial Rossby wave [12]; (ii) Hurricane Helene (2006) and an intensifying AEW [6]; (iii) Twin TCs (2002) and a mixed Rossby gravity wave [13] during an active phase of a Madden–Julian oscillation; and (vi) Hurricane Sandy (2012) [29] and upper-level tropical waves associated with a Madden–Julian oscillation [14]. The above studies collectively support the view that a large-scale system (e.g., tropical waves) can provide determinism on the prediction of TC genesis, making it possible to extend the lead time of genesis predictions. For example, analyses using traditional methods found that 30-day runs in [6] produced realistic initiation, as well as the propagation of six consecutive AEWs, between late August and late September 2006 and the mean state of an African easterly jet (AEJ) over Africa and downstream within the tropical Atlantic (Figure 1c,d). The results suggested a relationship between the improved simulations for an individual AEW as well as its interaction with local environments and a very impressive simulation of Helene's formation downstream from Day 22 to Day 30.

Since global simulations possess multiscale processes and interactions, the immediate question was how to reveal time varying scale interactions crucial for the formation of the above TCs? To address the above question, we developed tools for multiscale analysis and visualizations. In the multiscale analysis package, the core technology is empirical mode decomposition (EMD) and ensemble EMD (EEMD) methods. Both have shown remarkable performance in revealing multiscale processes of non-stationary and nonlinear data. The EMD method was originally developed by Huang et al. (1998) [30] and was extended to the EEMD by Wu and Huang (2009) [31] through the addition of ensemble members. The inclusion of ensemble computations into the EEMD was executed in order to overcome the scale (or mode) mixing problem of the EMD. Both methods decompose one set of observational data into so-called intrinsic mode functions (IMFs) that represent various oscillatory components of the data. One of the unique features of both methods is the possession of a filter bank property (e.g., [15,32,33]) (i.e., decomposed mean IMFs staying within natural filter period windows, to be discussed in Section 3.6). We implemented a three-level parallelism into the EEMD, referred to as the parallel EEMD (PEEMD), to improve parallel performance [15,34]. The newly developed PEEMD was first used to perform a multiscale analysis of Hurricane Sandy (2012) [15]. We then utilized it to perform a 10-year analysis on global reanalysis data (e.g., [16,35]), showing the statistics of cases with energy transfer between developing systems and environmental flows. To focus on the performance of the PEEMD in revealing the intensification of an AEW and its association with the formation of Hurricane Helene (2006), a brief summary is provided in Section 3.6. Such an analysis

provides additional support to the view that large scale processes provide determinism for the time evolution and spatial location of tropical cyclones.

The simulations of large-scale flows and mesoscale TCs were satisfactorily compared with the observations (e.g., reanalysis and best track). The aforementioned studies called for revisiting the predictability problem in order to understand whether or not such simulations were consistent with chaos theory. Specifically, we asked whether or not the so-called predictability limit of two weeks applied to this case. And if not, we asked why. Based on a comprehensive literature review and the development of a generalized Lorenz model, insightful understanding regarding chaos, butterfly effects, and predictability were obtained, as summarized in the following discussion. While the Lorenz 1963 model [1] was developed in order to determine the sensitive dependence of solutions on the initial conditions, which was then used to define "chaos", the Lorenz 1969 model [2] was proposed in order to reveal the instability of basic wind, showing the dependence of growth rates and predictability on spatial scales. Deterministic chaos within the Lorenz 1963 model suggested finite predictability, which was fundamentally different from the Laplacian view of deterministic predictability. By comparison, the appearance of instability within the Lorenz 1969 model indicated a finite practical predictability. The degree of chaotic responses displayed a dependence on system parameters (representing a system's heating or dissipation). The growth rates of solutions associated with system instability displayed dependence on the selection of basic winds (e.g., with a different slope for the wind spectrum, [36]) and physical processes (e.g., with or without dissipation, [37]) in the governing equations. Thus, an estimate of the predictability limit using the above simplified models should be interpreted with caution, as discussed in Sections 3.1–3.3.

Progress in the aforementioned global modeling has been enabled by recent advances in both supercomputing and visualization technologies over the past 15 years or so, in particular after the birth of the Japan Earth Simulator and NASA's Columbia supercomputer (e.g., [38,39]). High-resolution, high-fidelity global simulations of TCs and large-scale waves effectively revealed the role of hierarchical multiscale processes, as well as butterfly effects, in TC simulations. For example, a TC whose vertical structure varies with height possesses two major features, such as low-level cyclonic circulation and upper-level anticyclonic circulation. Such features of a TC and its interaction with large-scale environmental processes can better be displayed using quasi, three-dimensional streamline packages and concurrent visualization technology (e.g., [40,41]), which enable both high spatial and temporal resolution solutions. A summary of the technical details and visualizations are provided in Section 2.4, Section 3.3, and Section 3.4.

Sections 2.1 and 2.2 briefly introduce the global mesoscale model (GMM) and global reanalysis data, respectively. The PEEMD and its application for the multiscale analysis are discussed in Section 2.3. Section 2.4 applies 2D or 3D concurrent visualizations in order to reveal multiscale processes and initial noise within the model simulations. Section 2.5 presents the generalized Lorenz model (GLM). Section 3.1 discusses various types of solutions within the Lorenz 1963 model and the GLM, including nonlinear periodic solutions, also known as limit cycle solutions, and attractor coexistence. Section 3.2 provides a brief review of Lorenz 1969 study [2] for a comparison. Impact of errors of small scale processes and 30-day global simulations of large scale systems are discussed in Sections 3.3 and 3.4, respectively. Section 3.5 briefly comments the simulation of two hurricanes with larger errors. Section 3.6 summaries the 10-year analysis using the PEEMD. By comparing AEW simulations obtained using the GMM and limit cycle solutions of the GLM, Section 3.7 proposes a hypothesis for the recurrence (or periodicity) of AEWs during summer. Concluding remarks are provided at the end.

## 2. Numerical Model, Global Data, Visualization, and Analysis Methods

### 2.1. The Global Mesoscale Model

Enabled by advanced computational technologies, a global mesoscale model (GMM) at the highest horizontal resolution of 1/12 degrees (approximately 9 km) was deployed in 2005 [38], producing encouraging forecasts of intense hurricanes [6,10–12]. In this study, from a perspective of the generalized Lorenz model, we performed an analysis of global mesoscale simulations reported in earlier studies by myself and other colleagues. These simulations were obtained using the GMM that consists of three major components: finite-volume dynamics [42], the National Center for Atmospheric Research (NCAR) Community Climate Model physics, and the NCAR Community Land Model (CLM). Control runs were performed using basic model configurations including a 1/4 or 1/8 degree resolution and a large-scale condensation scheme. Such modeling settings enabled latent heat release from grid-scale condensation processes when cumulus parameterizations were disabled. The model configurations systematically produced reliable results. Our analysis indicated that a quasi-equilibrium assumption within cumulus parameterizations may limit the scale interaction between convection and large-scale flows and, thus, cause uncertainties in simulations of AEWs as well as hurricanes. Additional discussions regarding the impact of including or not including cumulus parameterizations in high-resolution simulations were provided in [10]. The best tracks for TCs were made available by the National Hurricane Center (NHC).

Additional parallel runs were performed using varying physics (e.g., different cumulus parameterizations) in order to understand the underlying dynamics and to examine the sensitivity of solutions to the initial conditions and/or physical processes. In this study, the control and three parallel runs, listed in Table 1, were further analyzed. Two parallel experiments containing different dynamic conditions during different months, referred to as experiments P1 and P2, respectively, were designed in order to reveal the dominant impact of land and physical processes. Experiments P1 and P2 used dynamic initial conditions (ICs) from 22 April and 22 June, respectively. To maintain the same model physics (e.g., radiation) and land model configurations as the control run, timestamps in the dynamic ICs were changed to 22 August. The goal was to show whether or not and how an AEJ that does not appear within the IC may be simulated using realistic land and physics conditions.

**Table 1.** Sensitivity experiments for studying the dependence of AEW simulations on different dynamic ICs and modified Guinea highlands. Initial conditions for physics and land processes remain the same in all experiments, as shown in the third column.

| Case ID | Dynamic IC | CLM and Physics IC | Guinea Highlands |
|---------|-----------|--------------------|------------------|
| CNTL | 08/22 | 08/22 | - |
| P1 | 04/22 | 08/22 | - |
| P2 | 06/22 | 08/22 | - |
| P3 | 08/22 | 08/22 | A factor of 0.6 in heights |

In the third experiment, P3, mountain heights were reduced in order to test whether or not and how lowering mountain heights (e.g., resolved in coarser resolution models) in the downstream impacts the simulation (and the successive initiation) of AEWs in the upstream. This was achieved as follows: within the longitude 16.5° W to 6° W and latitude 5° N to 13° N, mountain heights were multiplied by a factor of 0.6. The reduced mountains could have a "direct" impact on interactions with an approaching AEW, and an indirect impact that changed environmental flows and the subsequent initiation of AEWs upstream. Both could have an impact on a simulation of downstream hurricane formation.

### 2.2. Global Reanalysis Data

For this study, we selected the latest European Centre for Medium-Range Weather Forecasts (ECMWF) global reanalysis, i.e., European Reanalysis Interim (ERA-Interim) dataset [35]. The data

spans the time period from January 1979 to the present day. The dataset has a sampling rate of six hours and a horizontal grid spacing of 0.75 degrees, yielding a spatial resolution of approximately 78 km. For multiscale analysis using the PEEMD [16], zonal and meridian winds at 700 hPA were analyzed over a 10-year period from 2004 to 2013. A brief summary is provided below.

### 2.3. Parallel Ensemble Empirical Mode Decomposition (PEEMD)

As mentioned, the EEMD was developed to include sufficient ensemble members in order to obtain an ensemble average of IMFs, minimizing the issue of mode mixing. Depending on the required level of accuracy within the decomposed IMFs, 200–400 ensemble members are often used to obtain averaged IMFs, significantly increasing computational demands. Therefore, a three-level parallelism was implemented into the EEMD, referred to as the parallel EEMD (PEEMD) method, to efficiently and effectively reveal multiscale processes from high-resolution, global, multi-dimensional Earth science data [15]. The PEEMD achieved promising scalability with a parallel speedup and efficiency of 52.8 and 63 percent, respectively, by increasing the number of cores from 60 to 5000 [34]. In Section 3.6, we discuss a 10-year multiscale analysis using the PEEMD to reveal the scale interactions that contributed to the formation of Hurricane Helene (2006).

### 2.4. Streamline Package and Concurrent Visualizations

To visualize the multiscale interactions of a TC with its environmental flows, a time series of 2D frames of streamlines for TC circulations were produced at each vertical level, and the frames of several (e.g., 3–5) contiguous levels were grouped into one layer. The streamlines in the three different layers, low, middle and upper, are shown in blue, green, and pink, respectively. With other features such as opacity, used to control the (vertical) "transparency" of streamlines at different heights, the evolution of streamline density at a specific level qualitatively depicted the evolution of average wind speeds (i.e., denser streamlines signified stronger average wind speeds). See the technical details in [41]. To effectively process the simulations into visualizations, concurrent visualization technology was deployed and integrated within the model (e.g., [43]). Such an integrated system generated simulation data and, concurrently, produced animations. Specifically, when the model completed integration at each time step, the integrated system sent the model's outputs from the computing nodes to the visualization nodes via direct remote memory access, and a visualization module started processing the data and generating image frames for display. As a result, the simulations at ultra-high spatial and temporal resolutions were feasible. We showcase both 2D and 3D visualizations in Sections 3.3 and 3.4, respectively.

### 2.5. The Generalized Lorenz Model (GLM)

Over the past several years, a series of papers on high-dimensional Lorenz models [17–20] have yielded the following generalized Lorenz model (GLM) [22]:

$$\frac{dX}{d\tau} = \sigma Y - \sigma X \tag{1}$$

$$\frac{dY}{d\tau} = -XZ + rX - Y \tag{2}$$

$$\frac{dZ}{d\tau} = XY - XY_1 - bZ \tag{3}$$

$$\frac{dY_j}{d\tau} = jXZ_{j-1} - (j+1)XZ_j - d_{j-1}Y_j, \ j \in \mathbb{Z} \ : \ j \in [1, N] \tag{4}$$

$$\frac{dZ_j}{d\tau} = (j+1)XY_j - (j+1)XY_{j+1} - \beta_j Z_j, \ j \in \mathbb{Z} \ : \ j \in [1, N] \tag{5}$$

$$N = \frac{M-3}{2}; \; d_{j-1} = \frac{(2j+1)^2 + a^2}{1+a^2}; \; \beta_j = (j+1)^2 b \qquad (6)$$

Here, $\tau$ is dimensionless time. The three integers are $j$, $M$, and $N$. While $M$ represents the total number of modes (or equations), $N$ indicates the total number of pairs $(Y_j, Z_j)$ for higher wavenumber modes that do not appear within the original three-dimensional Lorenz Model (3DLM, [1]). The time-independent parameters include $\sigma$, $r$, $a$, $b$, $d_{j-1}$, and $\beta_j$. The first two represent the Prandtl number and the normalized Rayleigh number (or the heating parameter), respectively (e.g., [17]). The parameter a is defined as the ratio of the vertical scale of the convection cell to its horizontal scale, which is equal to $1/\sqrt{2}$. The last three are coefficients for the dissipative terms. Detailed discussions on each of the above terms can be found in [22]. The GLM with many modes was derived from the partial differential equations (PDEs) for Rayleigh Benard convection, while different chaotic systems with many modes proposed by Lorenz in 2005 [44] were not derived from physically-based PDEs. Variable X denotes the amplitude of the Fourier mode for stream function that defines horizontal and vertical velocities. The variables $(Y, Z)$, $(Y_1, Z_1)$, $(Y_2, Z_2)$, and $(Y_3, Z_3)$, referred to as the primary, secondary, tertiary, and quaternary modes, respectively, represent the amplitudes of the Fourier modes at different wavenumbers for temperature. When the nonlinear term $-XY_1$ is ignored, Equations (1)–(3) become the classical 3DLM. The results obtained from the GLM with $M = 5, 7$, and $9$, referred to as the 5DLM, 7DLM, and 9DLM, respectively, are presented in the following sections.

## 3. Discussion

### 3.1. New Insights into Predictability and Chaos

Chaotic solutions using the 3DLM have been a focal point and have led to the statement of "weather is chaotic". In fact, depending on the relative strength of heating, chaotic solutions appear as one of three types of solutions within the 3DLM, including:

1.   Steady-state solutions with small heating parameters (i.e., $r < r_c$; $r_c = 24.74$);
2.   Chaotic solutions with moderate heating parameters (i.e., $r_c < r < R_c$; $R_c = 313$);
3.   Limit cycle solutions with large heating parameters (i.e., $R_c < r$, e.g., [45,46]).

Additionally, the coexistence of chaotic and steady-state solutions may appear within a small range for the heating parameter, (i.e., $24.06 < r < 24.74$). In this paper, we briefly discuss the second and third types of solutions within the 3DLM and the coexistence of two types of solutions within the GLM.

We first present two features of chaotic solutions, including the divergence of nearby trajectories and solution boundedness. The sensitive dependence of solutions on ICs has been illustrated using the divergence of two initial nearby trajectories within the phase space of the 3DLM. For example, using the 3DLM with typical parameters (e.g., $\sigma = 10$, $b = 8/3$, and $r = 28$), Figure 2a–c displays a very different time evolution for two solution orbits whose starting points are very close to one another. In addition to the divergence of nearby trajectories, the solutions or orbits are bounded. Solution boundedness is indeed indicated by the finite size of the butterfly pattern for a chaotic solution. For such a system, the divergence of two orbits, which may be viewed as an error between a control and parallel run, should be bounded (e.g., Figure 2d).

When the Rayleigh parameter becomes large (say, $r > R_c$; $R_c = 313$), nonlinear oscillatory solutions appear (e.g., [45–47]). As show in Figure 3a, 200 runs with different ICs eventually approach the same closed orbit in black. Such a convergence of orbits indicates the isolated nature of a closed stable orbit where nearby trajectories approach the stable orbit. As a result of its isolated and closed nature (in Figure 3b), the nonlinear oscillatory stable orbit is indeed a limit cycle solution [48,49]. One interesting characteristic of a limit cycle is that its orbit is solely determined by the system and independent of the ICs. The "closed" feature associated with periodicity (or a "recurrence" feature with quasi-periodicity for a limit torus) can be illustrated using the non-dissipative model (e.g., [21,22,50]), suggesting that

nonlinearity alone or competition between heating and nonlinearity may produce nonlinear periodic or quasi-periodic solutions. By comparison, the isolated feature of limit cycles within the 3DLM indicates the importance of weak dissipation. An important message for the appearance of limit cycle solutions at large Rayleigh parameters is that chaotic solutions only occur over a finite range of Rayleigh parameters.

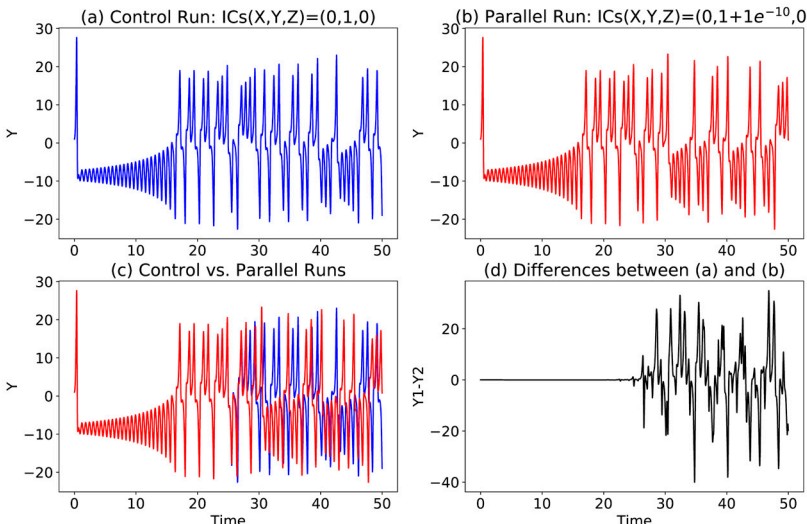

**Figure 2.** An illustration of the bounded divergence of two nearby trajectories within the 3DLM with $r = 28$ and $\sigma = 10$. Panels (**a**) and (**b**) display solutions from the control and parallel runs, respectively, the latter of which adds a small perturbation ($1 \times e^{-10}$) into the initial value of Y. Panel (**c**) reveals the sensitive dependence of solutions on the initial conditions. Panel (**d**) displays bounded differences in solutions for the control and parallel runs.

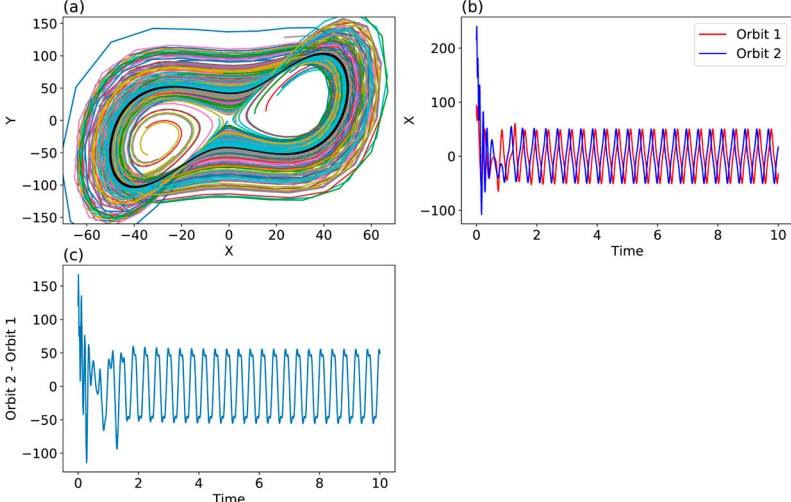

**Figure 3.** (**a**) A limit cycle (black) as indicated by the convergence of 200 orbits (color) beginning with 200 different initial conditions for $r = 350$ within the 3DLM. Color lines depict orbits during the period of $\tau = 1$ and $\tau = 10$ (i.e., $\tau \in [1, 10]$). "A thick black line" displays 200 orbits for $\tau \in [9, 10]$. Convergence of the 200 orbits into the black orbit indicates the isolated nature of a limit cycle. Panels (**b**) and (**c**) display two orbits and their differences, respectively.

Compared to the 3DLM, the GLM, with any odd number of M greater than three, possesses the following features: (1) three types of solutions (that also appear within the 3DLM); (2) two kinds of attractor coexistence; (3) aggregated negative feedback; and (4) hierarchical scale dependence. Below we briefly discuss (3) and (4) and then illustrate (2). Within the GLM, the negative feedback

associated with smaller-scale modes can be aggregated to provide a stronger effective dissipation. As such, a higher critical value for Rayleigh parameter is required for the onset of chaos in a higher dimensional Lorenz mode. This can be seen in Table 2. As indicated by the high Pearson correlation coefficients between the primary and secondary modes ($Z$ and $Z_1$) and between the secondary and tertiary modes ($Z_1$ and $Z_2$), 0.988 and 0.998, respectively, Figure 4 using the 7DLM displays hierarchical scale dependence within the chaotic solutions.

**Table 2.** The characteristics of various Lorenz models. Values for $r_c$ are determined based on analyses of the ensemble Lyapunov exponents [17,51]. The "Heating terms" column indicates heating terms within the corresponding LM.

| Model | $r_c$ | Heating Terms | References |
|:---:|:---:|:---:|:---:|
| 3DLM | 23.7 | $rX$ | [1] |
| 5DLM | 42.9 | $rX$ | [17] |
| 6DLM | 41.1 | $rX, rX_1$ | [18] |
| 7DLM | 116.9 | $rX$ | [19] |
| 8DLM | 103.4 | $rX, rX_1$ | [20] |
| 9DLM | 102.9 | $rX, rX_1, rX_2$ | [20] |
| 9DLM (new) | 679.8 | $rX$ | [22] |

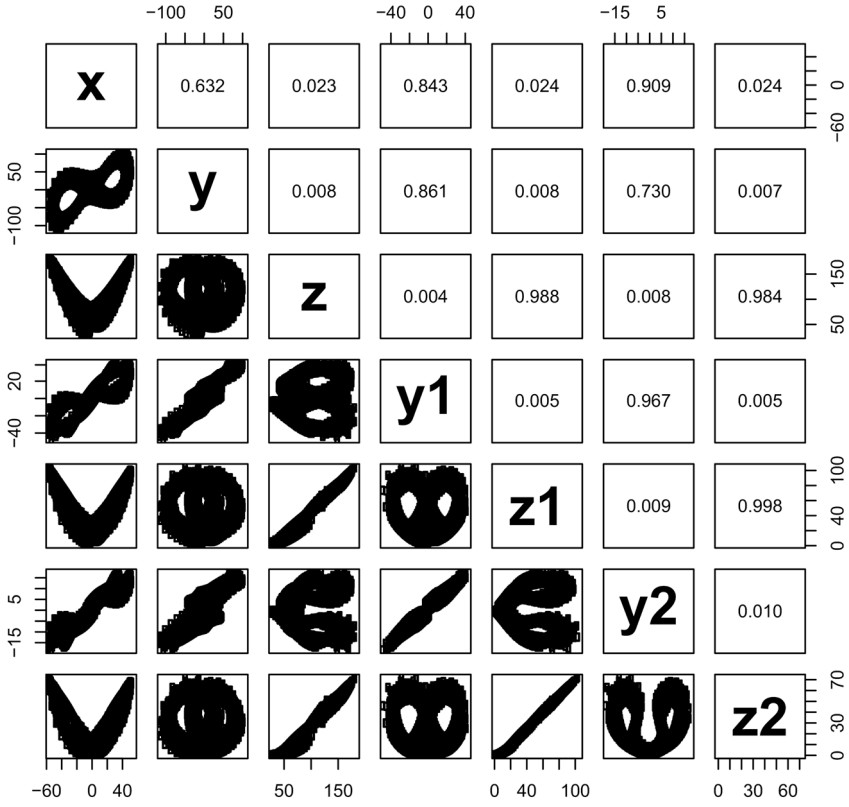

**Figure 4.** A matrix of scatter plots for the 7DLM with its seven variables, as listed in the principal diagonal. *r* and *σ* are 120 and 10, respectively. Each of the cells above (below) the diagonal provides a Pearson correlation coefficient (scatter plot) between the two variables. Scale dependence is indicated by high Pearson correlation coefficients, as well as the linear relationship in scatter plots (courtesy of [20]; See details in [19]).

Here, two kinds of attractor coexistence are discussed. The first kind of attractor coexistence, with chaotic and steady-state solutions, appears in a wider range of Rayleigh parameters in higher-dimensional Lorenz models (i.e., $679.8 < r < 1058$ for the 9DLM) as compared to a smaller range of Rayleigh parameters within the 3DLM. The appearance of either chaotic or non-chaotic solutions depends on the initial conditions. Within the 3DLM or 9DLM, coexisting attractors appear within a system that has stable, non-trivial equilibrium points ([22] and references therein). Therefore, it is reasonable to hypothesize that orbits beginning "near" the stable non-trivial critical points may move toward the non-trivial equilibrium point, at least from a statistical perspective. To show such features using the 9DLM, a large number of runs using different initial conditions were performed. A Gaussian random generator was applied in order to produce $N$ ($N = 256$) data points that were distributed over a hypersphere centered at the non-trivial equilibrium point with a radius of $R$ ($R = 300$). As shown in Figure 5 with $r = 680$, the 256 orbits could be classified as chaotic orbits and steady-state solutions, indicating the coexistence of two types of solutions. A detailed analysis of the dependence of chaotic and non-chaotic solutions on ICs using the ensemble modeling approach with various Ns and Rs can be found in Figure 5 of [23]. The second kind of attractor coexistence with limit cycle and steady solutions was first documented by Shen [22]. Such coexistence was further discussed using the 9DLM with $r = 1600$ and 128 different ICs in Figure 9 of [52]. From a practical perspective, better predictability can be obtained for non-chaotic solutions with insensitivity to the initial conditions.

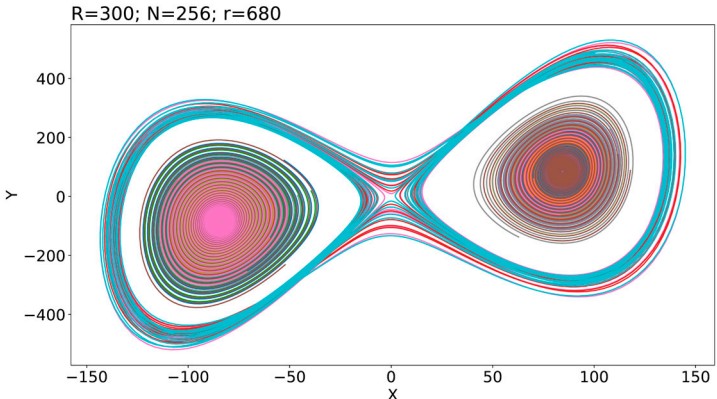

**Figure 5.** The coexistence of chaotic and non-chaotic orbits within the GLM with $M = 9$; 256 ICs distributed over a hypersphere with a radius of 300. $\tau \in [0.625, 5]$ with $\Delta\tau = 0.001$.

As a result of the above four features within the GLM, a refined view of the nature of weather is proposed [22,23] as follows: The entirety of weather is a superset that consists of both chaotic and non-chaotic processes. In other words, both chaos and order may coexist. Such coexistence suggests better predictability for non-chaotic processes if we can identify them in advance (e.g., [52]).

### 3.2. A Brief Review of Lorenz (1969)

In addition to the 3DLM, Lorenz proposed another model with 21 modes in 1969 [2], showing two major features, the dependence of growth rates on scales and energy transferring across scales. The 3DLM and the 1969 model were used to reveal finite intrinsic predictability for chaotic solutions and finite practical predictability for unstable solutions, respectively. From the perspective of predictability estimates, the calculation of growth rates was more feasible in the 1969 multiscale, linear model than in the 1963 chaotic model since the latter displays sensitivity of solutions to the initial conditions and, thus, produces larger variations of time dependent growth rates. However, as a result of the simplicity in the dynamics and physics of these models as well as other existing models, any quantitative estimate of the predictability should not be viewed as an upper or lower bound of intrinsic predictability.

Lorenz (1969) [2] studied energy transferring across scales and its impact on estimates of growth rates using two experiments illustrating the impact of upscale and downscale transfer for initial error.

While upscale transfer had been a main focus in [2], the two experiments indeed produced comparable results with similar predictability over a range of scales (e.g., Table 3 of [2]). To this end, a recent study in [53] emphasized the role of the downscale transfer of an initial error, originally at a larger (synoptic- or meso-) scale, in producing a reduction of predictability on a smaller scale. These authors further suggested that the upscale transfer of error associated with "the butterfly effect" may not be so crucial for obtaining daily weather prediction (e.g., [54]).

Upscale transfer is a major feature in turbulence models. However, it is not clear whether the 1969 model is a turbulence model because the a conservative partial differential equation with a realistic (turbulent) basic state was used derived the 1969 model. By comparison, as documented on page 139 of Lorenz (1993) [55], the equations of the 3DLM lacked important properties associated with turbulence. Therefore, the 3DLM was not capable of addressing "deterministic turbulence" but illustrated "deterministic non-periodic flow". As a result, detailed similarities and differences in the underlying mechanisms for finite predictability within the 3DLM and the 1969 model of Lorenz should be examined to understand the role of upscale transfer. This will be undertaken in future studies.

### 3.3. Impact of Errors on Small Scale Processes

Based on the major findings of Lorenz's studies, the following two features have been accepted within the scientific community:

1.    Small-scale processes are less predictable than large-scale processes.
2.    Errors associated with small-scale processes may "quickly" contaminate simulations of large-scale flows.

Since small errors can easily appear in initial conditions, the above leads to a pessimistic view for extending prediction lead times beyond 5–7 days. For example, such errors appear in my colleagues and my modeling approaches that apply coarser-resolution National Centers for Environmental Prediction (NCEP) analysis data to drive higher-resolution model runs. However, as reported above, encouraging results were still obtained. Why? Here, an analysis on the above two features is provided. While the first feature may be derived from the result of scale dependence of growth rates on scales in [2], the second feature seems to be associated with the sensitive dependence of solutions on the initial conditions in [1]. For the Lorenz (1969) study [2], the model should have included realistic dissipations to allow dissipative small-scale processes with negative growth rates. For the Lorenz 1963 model [1], as well as for the GLM, the coexistence of two types of solutions suggests that a steady-state solution with a negative growth rate may coexist within a chaotic solution. Small-scale processes with negative growth rates are stable and possess deterministic predictability. As a result, the first statement is not very accurate. The coexistence of chaotic and steady-state solutions suggests that tiny errors associated with chaotic solutions should not cause a large impact on steady-state solutions, and vice versa. The second statement is also not accurate. This is qualitatively illustrated in Figure 6 and its corresponding animation for the global simulation of vertical velocity over an initial 5-day period. The animation shows that some noise associated with initial imbalance between the higher-resolution GMM and coarser-resolution NCEP reanalysis data were dampened, yielding no significant impact on the subsequent simulations.

In summary, the two features outlined in the beginning of this subsection may not always occur. Although dissipative small-scale processes should be predictable and may not necessarily contaminate the simulations of large-scale flows, numerical models that contain highly dissipative processes may have a numerical issue with the so-called stiff problem, which can be illustrated using the simplest ODE: $dy/d\tau = \lambda y$, with a negative $\lambda$ but a large $|\lambda|$ [56]. Note that mathematically and physically, the solution is stable with a large decay rate. However, numerical instability may appear within such a system, thereby requiring a better scheme or a very small time step in order to improve solution stability. For global high-resolution models with many scales, a stiff problem may appear as a result of a large ratio between the largest and smallest eigenvalues and, thus, produce unstable solutions.

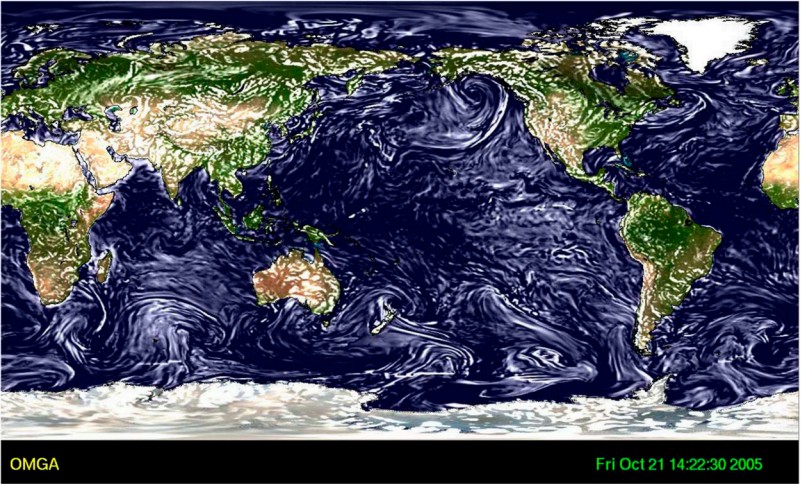

**Figure 6.** A selected frame from a global animation of the vertical velocity in pressure coordinates from a run initialized at 0000 UTC 21 October 2005. The corresponding animation is available as a google document: http://bit.ly/2GS2flD. The animation displays dissipation of the initial noise associated with an imbalance between the model and the initial conditions.

### 3.4. 30-Day Global Simulations of Large-Scale Systems

We previously discussed a remarkable 30-day simulation with a realistic formation simulation for Hurricane Helene (2006). Hurricane Helene was found to be related to interactions between an observed AEW and the local environment (e.g., [57]). Below we provide a brief summary on simulations of AEWs and an AEJ during the 30-day period. In our study the criteria for the analysis of AEW simulations was similar to those in [58]. The ridge (or the trough) of the AEW was defined as the location with a transition from southerly to northerly flow (or from northerly to southerly flow).

Time–longitude diagrams of 850 hPa meridional winds averaged over 5° to 20° N are shown in Figure 1c. The figure displays the occurrence of six westward propagating AEWs over the 30-day period. The waves had a timescale of 3–5 days, a wavelength of approximately 2000–2500 km, and a propagation speed of roughly 10 m/s. Overall, the model simulations of multiple AEWs were in good agreement with the analysis, especially over the African continent. Spatial and temporal variations existed but were within one characteristic (time and spatial) scale. The strong wind shear along 20° W during 11–13 September, as shown by the black circle in Figure 1c, roughly indicated the formation of Hurricane Helene.

Additionally, Figure 1d shows altitude–latitude cross-sections of zonal winds averaged over the 30-day period along longitude 10° E from the control run. A low-level jet with a maximum of around 10–14 m/s at (14° N, 600 hPa), referred to as the AEJ ([59]), was clearly evident. Overall, the model simulation was in good agreement with the NCEP analysis, but the simulated AEJ was slightly weaker. Below and south of the AEJ, a low-level westerly monsoon flow was simulated (Figure 1d). At roughly 200 hPa and equatorward of the AEJ, the model simulated an upper-level tropical easterly jet that appeared at the right altitude but had a stronger intensity between 9° N and 15° N, as compared to analysis data.

Promising simulations with multiple AEWs and an averaged AEJ support the view that large-scale processes may determine modulation within the simulation of Hurricane Helene (in Figure 1a,b). As shown in Figure 7, such a feature can be illustrated using a quasi, three-dimensional streamline visualization that depicts the evolution of an intensifying AEW into a hurricane from a 30-day simulation.

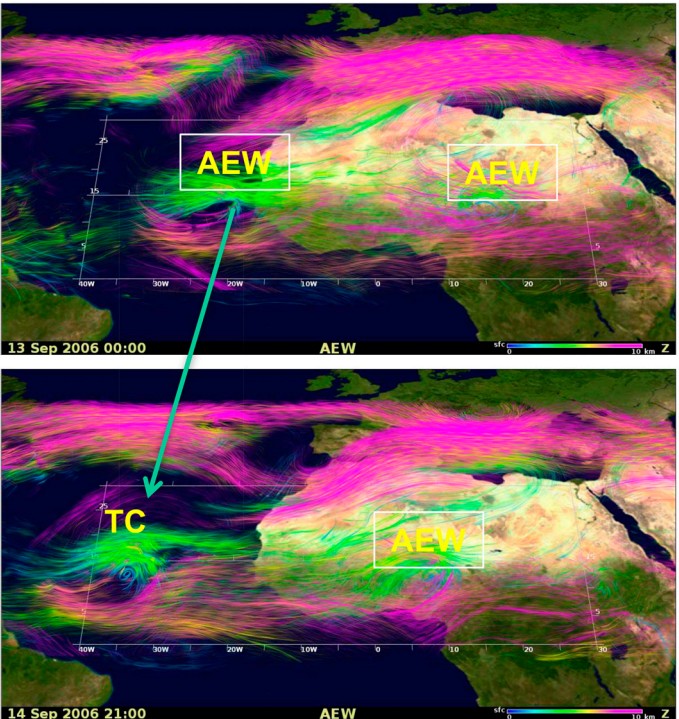

**Figure 7.** A visualization of formation of Hurricane Helene (2006) and its association with the intensification of an African Easterly Wave (AEW) in a 30-day run initialized at 0000 UTC 22 August 2006. Upper-level winds are shown in pink, middle-level winds in green, and low-level winds in blue. (**top**) Initial formation of Helene as the AEW moves into the ocean, validated at 0000 UTC 13 September 2006 (day 22); (**bottom**) initial intensification associated with intensified low-level inflow with counter clockwise circulation, validated at 2100 UTC 14 September 2006. An animation can be found at: http://tiny.cc/j9ul9 (courtesy of [41]).

### 3.5. Simulations of Hurricanes Debbie and Florence

In addition to Hurricane Helene, the 30-day control run in [6] also simulated two other hurricanes, Hurricanes Debbie and Florence, for their movement and formation, respectively (as shown in Figure S6 of the supplemental materials in [6]. The control run did not produce any false alarm. Overall, the movement of hurricane Debbie, which appeared in the ICs, was well simulated. Simulated Hurricane Florence first appeared 10° east of the observed hurricane several days earlier, yielding larger errors in location and timing. Different accuracy in the formation simulation for Hurricanes Florence and Helene may be explained as follows. While the model possesses two-way interaction between the surface and atmosphere over land (provided by the land model), it only contained one-way interaction over the ocean (provided by the prescribed weekly sea surface temperature). As a result, the predictability of hurricane formation may be better near the Cape Verde Islands that are closer to the continent (e.g., for Helene) than over the Atlantic Ocean (e.g., for Florence).

### 3.6. Downscaling Processes Revealed by the PEEMD

Here, we first illustrate that the EMD/EEMD/PEEMD behave as a collection of band-pass filters [32,33] and, thus, are ideal for multiscale analysis. We previously generated a data set with one million points as Gaussian white noise and decomposed the data set into nine IMFs using the EMD. For each of IMFs a spectrum was determined using Fourier (spectral) analysis. As shown in Figure 8, the spectrum of each IMF that displays a Gaussian distribution contained signals within a range of frequencies. In general, a higher order IMF has lower frequencies (or wavenumbers) associated with larger temporal (or spatial) scales. Specifically, the averaged period (or wavelength) of the $(n + 1)$th

IMF doubles the period (or wavelength) of *n*th IMF (e.g., [15]). This feature, similar to that of a dyadic filter, is ideal for multiscale analysis.

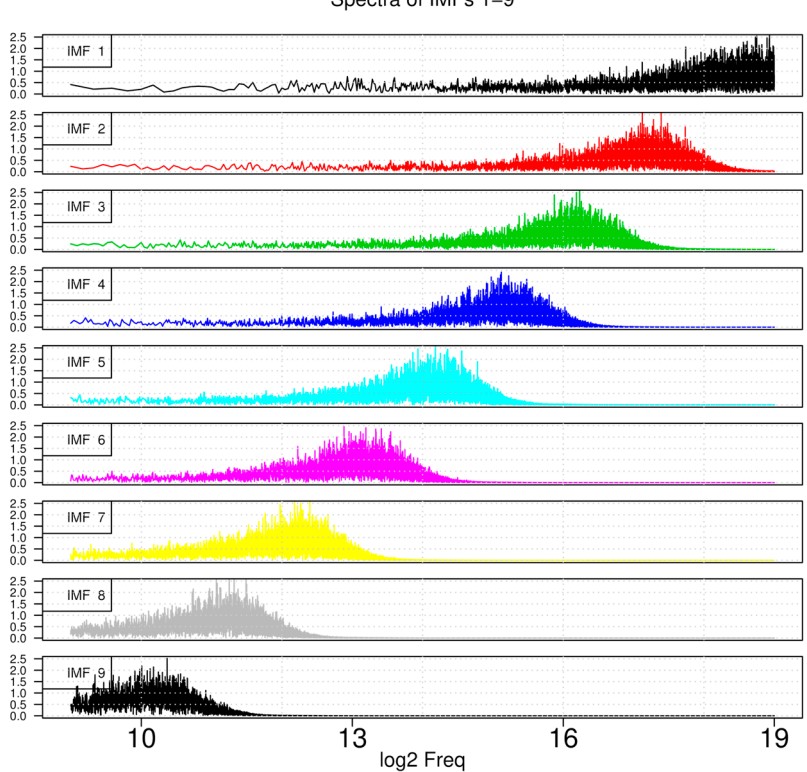

**Figure 8.** The feature of a collection of band-pass filters within the EMD/EEMD/PEEMD. Panels display the spectra of the first nine intrinsic mode functions (IMFs) that were obtained from the decomposition of Gaussian white noise with one million points. The horizontal represents the logarithm of the frequency (courtesy of [15]).

The PEEMD was applied in order to extract oscillatory IMFs from ECMWF global data. The difference between the raw data and a sum of its oscillatory IMFs is defined as a residual, called a trend mode. In the 10-year data analysis, we found that the trend mode and the third IMF (IMF3) largely represented environmental flows and a mixed AEW and TC, respectively. Below we illustrate energy transfer between the trend mode and IMF3 during the intensification of an AEW and the formation of a TC, including Hurricane Helene (2006). Table 3 summarizes an analysis of the ERA-interim dataset and the NHC best tracks dataset from July 2004–September 2013. In the data, the number of AEWs remained nearly constant and the number of TCs changed significantly from year to year. The former seemed to suggest "stable" large-scale forcing (e.g., seasonal forcing remained nearly unchanged inter-annually), while the latter suggested various mechanisms for the further intensification of an AEW that may or may not have developed into a TC. Since these characteristics were observed during July, August, and September, they may indicate the importance of low-level heating/radiation.

For developing cases, as shown in Figure 9 for Hurricane Helene, the time evolution of wind shear along their tracks was analyzed to reveal downscaling processes during intensification. Storm intensification was indicated by the dropping of sea-level pressure at the storm's center. During storm intensification, the shear of the trend mode decreased with time, although such a tendency was not clear in the shear of total wind. At the same time, the shear of the IMF3 mode gained strength. As a result, the decrease and increase of shear for the trend and IMF3 modes, respectively, indicated the role of downscaling transfer from the trend mode to the IMF3 mode in storm intensification. Below we provide statistics for cases with similar features of downscaling processes.

**Table 3.** Breakdown by year of AEWs (2nd column), the NHC tracked storms (3rd column), and hurricanes (4th column) for July, August and September from 2004 to 2013 for the main development region. In the 3rd column, numbers outside (or between) the parentheses include tropical depressions (TDs) that were (or were not) associated with AEWs. Numbers in the 4th column contain hurricanes that developed from the AEW associated storms and the numbers between the parentheses are hurricanes with downscaling features. Table was reproduced by permission of the American Meteorological Society [16].

| Year | No. of AEWs | No. of TDs | No. of Hurricanes |
|---|---|---|---|
| 2004 | 28 | 7(8) | 4(2) |
| 2005 | 28 | 3(6) | 2(1) |
| 2006 | 26 | 3(4) | 1(1) |
| 2007 | 28 | 4(5) | 3(1) |
| 2008 | 28 | 3(4) | 2(0) |
| 2009 | 30 | 4*(5) | 1(1) |
| 2010 | 27 | 6(8) | 4(2) |
| 2011 | 27 | 4(4) | 3(2) |
| 2012 | 27 | 5(7) | 4(2) |
| 2013 | 24 | 3(4) | 1(1) |
| total | 272 | 42(56) | 25(13) |

* The number includes one storm that was already a tropical system (TS) when first classified by the NHC.

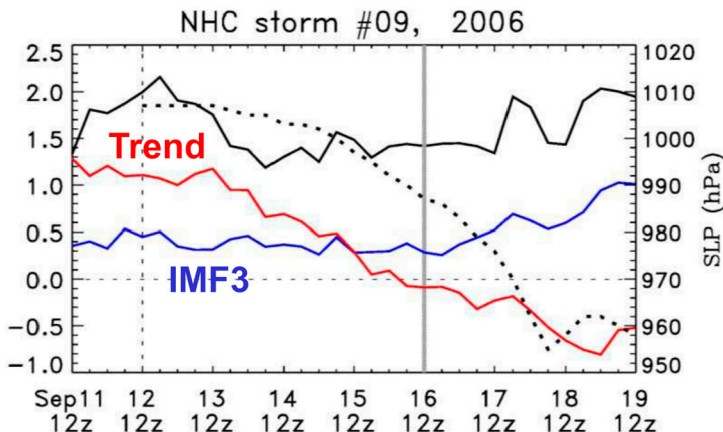

**Figure 9.** The tendency for horizontal wind shear of the total winds (black line), IMF3 (blue line), and the trend mode (red line) and the minimum sea level pressure (black dotted line) along the track of Helene (2006). The vertical thin dotted line indicates the time when the storm was first classified as a TD by the NHC, while the vertical gray line is provided at 1200 UTC 16 September 2006 as a reference line. The plotted shear of the IMF3 and the trend mode were multiplied by three. Figure was reproduced by permission of the American Meteorological Society [16].

With the goal of understanding the impact of AEWs on the formation of Cape Verde hurricanes, a selected domain for the analysis covered latitudes between 7° and 20° N and longitudes between 15° E and 60° W, the same main development region studied in [60]. Over the 10-year period, 42 AEWs developed into NHC classified storms within the selected domain and 25 further developed into hurricanes (Table 3). Decomposed components for the trend mode and IMF3 suggested that 13 of the 42 developing AEWs exhibited a downscaling feature for shear transfer from the trend mode to the IMF3. All of the 13 cases developed into hurricanes. In other words, 13 of 25 hurricanes were associated with prominent downscaling processes. For these 13 cases, the average drop of minimum sea level pressure

was 53 hPa during the intensification phase. The average decrease of the u-component wind shear for the trend mode, representing a decrease in the basic state wind shear, was $0.37 \times 10^{-5}$ s$^{-1}$, while the average enhancement of the u-component wind shear (as well as the v-component wind shear) of the IMF3 was $0.28 \times 10^{-5}$ s$^{-1}$ during storm intensification.

### 3.7. A Hypothetical Mechanism for Recurrence and Periodicity of Multiple AEWs

Here, the focus of the analyses is on the simulation and consecutive initiation of multiple AEWs. We formally introduce "recurrence" that is defined when the trajectory of a state returns back to the neighborhood of a previously visited state. Thus, recurrence braces quasi-periodicity and chaos and may be viewed as a generalization of periodicity [61]. The recurrence was clearly shown by the consecutive appearance of multiple AEWs (in Figure 1c). For example, Table 3 indicates 27 AEWs per 92 days (for July, August and September) each year. The "recurrence" time is about 92/27 = 3.4 days. The following discussions suggest that the recurrence may contribute to the predictability at extended-range (15–30 days) scales (in Figure 1a,b). Below we first provide a brief review on the key role of land surface processes in contributing to the predictability of both AEWs and the AEJ in extended-range simulations, a feature that can be viewed as a boundary value problem. As a result, extended-range simulations for these AEWs could be the mixed form of an initial value problem and a boundary value (forced) problem. Furthermore, we apply the dynamics of the limit cycle to propose a hypothetical mechanism for the recurrence (or periodicity) of AEWs.

In addition to Figure 1c, the nature of temporal "periodicity" could be qualitatively seen in Figure 10, displaying oscillatory correlation coefficients for the 850 hPa temperatures from the control run and the NCEP analysis over the 30-day period. Results with correlation coefficients above 0.65 suggest that temperatures are simulated with some degree of realism. Such results indicate the advantage of a high-resolution global model that can reduce uncertainties associated with imposed lateral boundary conditions, making it possible to perform longer simulations, as compared to regional models. However, the correlation coefficients oscillate with time. Why? Was this consistent with the divergent feature of chaotic processes? Are correlation coefficients supposed to be a monotonically decreasing function with time? We address this question below.

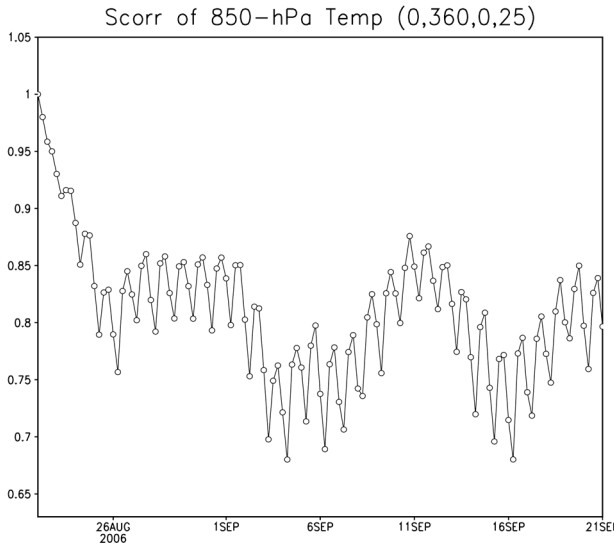

**Figure 10.** Correlation coefficients between the simulated 850 hPa temperature and the corresponding NCEP analysis over the domain longitude 0° E to 360° E and latitude 0° N to 25° N over the 30-day period. The choice of a global belt domain for verification is due to zonally-moving weather systems. Correlation coefficients are calculated with the scorr function provided by the Grid Analysis and Display System (GrADS). It should be noted that the correlation coefficients for the 850 hPa temperatures above are 0.65 for the entire 30-day period. Data are from the supplemental materials in [6].

In the early study [6], we additionally presented good agreement of the spatial distribution of 30-day averaged 850 hPa temperatures between the control run and the NCEP analysis (e.g., Figure 2 in [6]). The result may suggest the importance of surface forcing (or low-level forcing) on simulations of AEWs as well as hurricanes during the July–September period. As shown in Figure 11a,b via dynamic ICs on 22 April and 22 June, respectively, experiments P1 and P2 with no or a weaker AEJ initially indicated that a realistic AEJ and westward moving AEWs could be simulated using the same model physics (e.g., radiation) and land model configurations as the control run. For example, after 20–25 days of integration, experiment P1 was able to produce AEWs (Figure 11a) and an AEJ (Figure 11c–f), although simulated AEWs and AEJs had large errors. In the two experiments with very different dynamic ICs, large discrepancies in timing and location also existed. However, the two experiments indeed illustrated the importance of accurate land surface and physics ICs on simulations of AEW(s), and a realistic AEJ. Additionally, experiment P3 with reduced mountain heights was used to examine their impacts on AEW simulations, displaying higher impact on the downstream development of AEWs than on upstream initiation. This suggested that it takes time for errors to have impact on simulations.

Here a brief summary of the limit cycle solution is provided as a baseline for proposing a mechanism for the recurrence of the multiple AEWs. The GLM and 3DLM were derived based on partial differential equations for Rayleigh–Benard convection with heating at the bottom. The appearance of nonlinear limit cycle solutions at large Raleigh parameters suggested that: (1) the collective impact of nonlinearity and "strong" heating could lead to periodicity while (2) relatively small dissipation was responsible for the isolated nature of the limit cycle solution. Such an isolated feature was shown in Figure 3a, suggesting no long-term memory of initial conditions. In other words, a limit cycle solution is insensitive to tiny changes in initial conditions. Specifically, a limit cycle and its periodicity and amplitudes are solely determined by the system and thus independent of initial conditions within the idealized 3DLM and GLM that contain constant forcing and dissipative terms. An accurate IC is effective in helping reach the balance between the nonlinearity and heating, playing a role in determining the initial evolution of solution. Based on the analyses above and below, we propose that the periodicity of AEWs may be largely determined by strong surface heating and nonlinearity. The hypothesis was consistent with the fact that AEWs mainly appear during July, August, and September. Since periodic signals produced oscillatory differences between the control and parallel runs, as shown in Figure 3c for the limit cycle solution, the hypothesis was also consistent with the simulated result containing oscillatory forecast scores in Figure 10. Note that, by comparison, chaotic solutions produced irregularly oscillatory "errors" (or divergences) (Figure 2d).

The above discussions suggest the possibility of extending the lead time of prediction at extended-range scales (say to 23 days). For example, when a model has accurate land and physics components, as well as ICs, to simulate the "periodic" nature of AEWs with a period of 5 (or 3.4) days and has the capability of simulating the downscaling processes associated with the fourth (or fifth) AEW for a period of 3 days, it may produce simulations of TCs with a predictability of 23 (or 20) days, computed as follows: $5 \times 4 + 3 = 23$ (or $3.4 \times 5 + 3 = 20$). The 3-day predictability of downscaling processes is not inconsistent with findings using current regional models. As a result of the periodic (or recurrent) nature of large-scale systems (that lead to a high intrinsic predictability), extended-range predictability is possible.

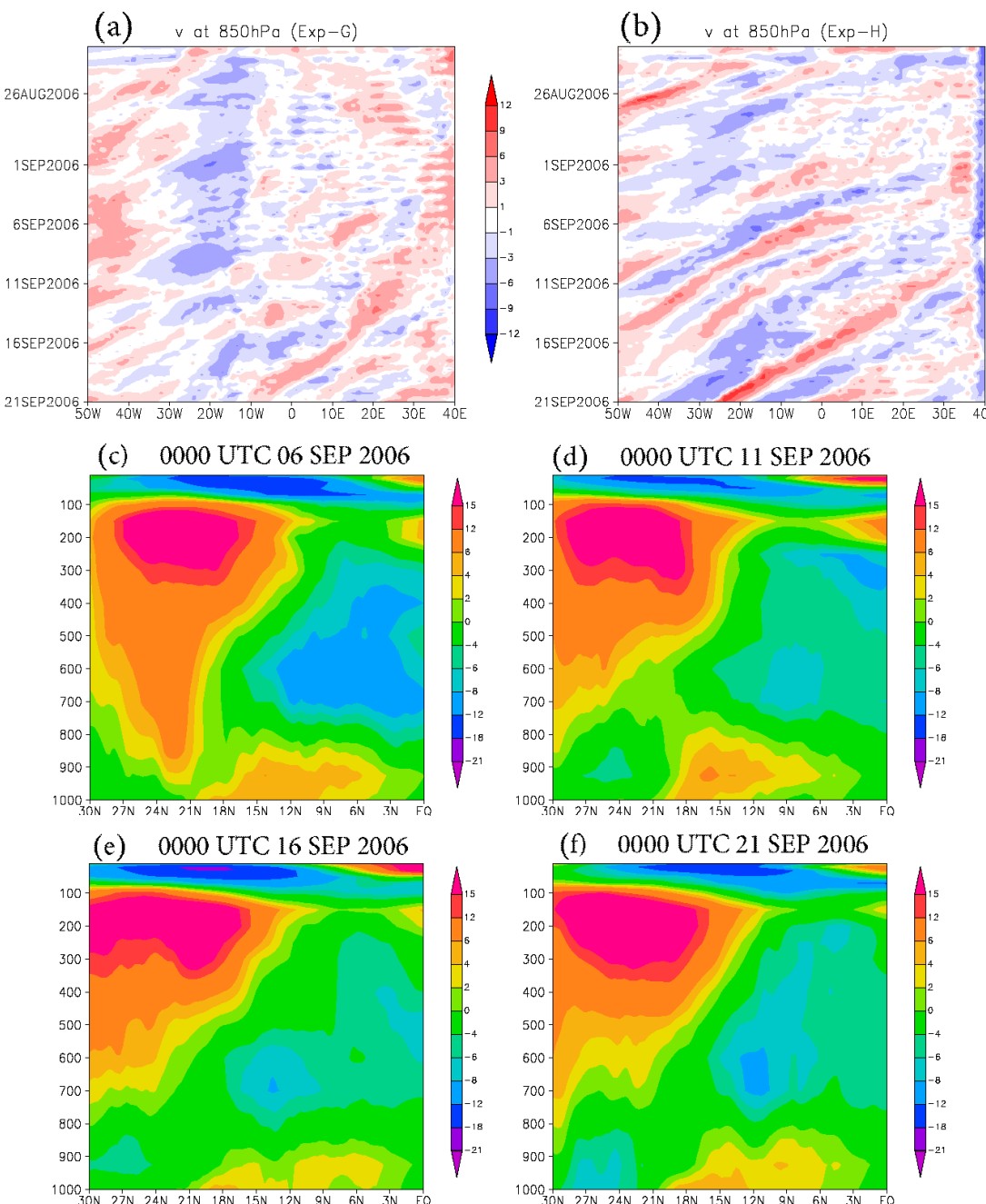

**Figure 11.** Panels (**a**,**b**) illustrate the sensitivity of AEW simulations to different dynamic ICs in experiments P1 and P2, respectively. Time-longitude diagrams of simulated meridional winds are plotted. Panels (**c**–**f**) display altitude–latitude cross sections of zonal winds averaged over longitude 20° W to 20° E on 6, 11, 16, and 21 September 2006, respectively, showing the development of an AEJ after 25 days in experiment P1. Data are from the supplemental materials in [6].

## 4. Conclusions

Recent advances in computational and global modeling technology have shown the potential for improving weather predictions at extended-range scales. In some of our past work, establishing remarkable predictability, we proposed a conceptual model that examined the role of (i) the downscaling processes of large-scale tropical systems (e.g., African easterly waves (AEWs) and Madden–Julian oscillations), and (ii) the upscaling processes of small-scale flows (e.g., precipitation) in the formation, intensification, and movement of mesoscale tropical cyclones (TCs). In earlier studies [6], we reported

realistic simulations of multiple AEWs and an averaged AEJ in 30-day runs and promising simulations of formation for hurricane Helene (2006) from Day 22 to Day 30. In this study, such extended predictability was further analyzed based on a recent understanding of chaos and instability derived from studies using the Lorenz models [1–3,25] and the generalized Lorenz model [22].

By definition, intrinsic predictability and practical predictability are different. Since numerical models cannot perfectly represent weather, an estimate of practical predictability should be interpreted with caution. While the Lorenz 1963 and 1969 models [1,2] suggest finite predictability, the underlining mechanisms are different. The former focuses on chaos (i.e., sensitive dependence on initial conditions) and the latter associates finite predictability with instability. Both models possess simplified physical processes (i.e., without realistic dissipations), so, in practice, an estimate of predictability using the above models may be a qualitative indicator but not necessarily an upper bound for intrinsic predictability. Therefore, the statement regarding a theoretical predictability of two weeks is not universal. The potential for extending the lead time of predictions was discussed by providing insight into understandings of chaotic and non-chaotic processes within the generalized Lorenz model (GLM).

The GLM possesses the following features: (1) three types of attractors (that also appear within the original Lorenz model), (2) two kinds of attractor coexistence, (3) aggregated negative feedback, and (4) hierarchical scale dependence. When additional realistic dissipative processes at smaller scales are included within the GLM, their negative feedback can be aggregated to produce a stronger dissipation to stabilize the system, leading to stable equilibrium points and steady-state solutions. The feature of hierarchical scale dependence provides a theoretical basis for the role of large-scale processes in modulating small scale processes. All of these features suggest a refined view on the nature of weather as follows: the entirety of weather is a superset that consists of chaotic and non-chaotic processes. Stable, steady-state solutions should have better predictability compared to chaotic solutions. Additionally, errors associated with dissipative, small-scale processes do not necessarily contaminate the simulations of large-scale processes.

In Lorenz systems, chaotic solutions may appear within a subset of the entirety of solutions. In addition to chaotic and steady-state solutions, nonlinear periodic solutions may appear alone in the 3DLM at large Rayleigh parameters or they may coexist with steady-state solutions in the GLM with M = 9 or larger. Based on a simple comparison between Rayleigh–Benard convection and AEW/AEJ problems, we applied the dynamics of the limit cycle solution in order to propose a hypothetical mechanism for the recurrence (or periodicity) of successive AEWs. The recurrence may appear as a result of a balance between the dominant surface heating and nonlinearity. Specifically, a system with strong heating balanced by nonlinearity may produce recurrence and, thus, be more predictable. The characteristics of recurrent signals (e.g., periods) may be less sensitive to initial conditions (as shown in Figure 11), as suggested by the insensitivity of limit cycles to initial conditions. By comparison, the initial evolution and the phase of oscillatory solutions may be influenced by accurate initial conditions. Such an impact by initial conditions can be seen in the simulations of two limit cycle solutions (Figure 3b) as well as in global model simulations of AEWs from the control run and parallel experiments P1 and P2. Therefore, within simulations of oscillatory (or recurrent) signals, forecast scores may be oscillatory. As a result of better predictability for recurrence, near the end of Section 3.7 we discussed the possibility of extending the lead time of predictions at extended range scales. Future work will include refining the idealized Lorenz models (e.g., with realistic parameters) to better examine the validity of the mechanism in explaining the recurrence of multiple AEWs.

**Funding:** This research received no external funding.

**Acknowledgments:** We thank reviewers and R. Atlas, R. Anthes, J.-J. Baik, D. Durran, F.D. Marks, Z. Musielak, T. Krishnamurti, C.-D. Lin, R. Rotunno, I. A. Santos, R. Pielke, Sr., X. Zeng, and F. Zhang for valuable comments and discussions. We are grateful for the support from the College of Science at San Diego State University. We thank J. Cui for his help in producing the figures. This paper was completed based on a recent presentation entitled "Butterfly Effects and Chaos within a Generalized Lorenz Model: New Insights and Opportunities" at NOAA/AOML/HRD http://bit.ly/2LuiIAY.

**Conflicts of Interest:** The author declares no conflict of interest.

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
