# Peer review of "On the Predictability of 30-Day Global Mesoscale Simulations of African Easterly Waves during Summer 2006: A View with the Generalized Lorenz Model"

_geosciences, doi:10.3390/geosciences9070281_

Reviewer 1 Report

This is a paper on the possibility to extend the theoretical predictability limit of two weeks in NWP to longer times. The argument rests on studies of a generalized Lorenz model.

The paper is interesting, fairly well written and the author is knowledgeable in the field. I recommend publication, provided the following comments have been responded to in the paper.

General comments:

1) It is claimed that “the entirety of weather is a superset that consists of chaotic and non-chaotic processes”. This is hardly news for the meteorologist. What is new, however, is the claim that for certain combinations of these processes, NWP is possible beyond two weeks. The author should be more explicit as to which cases, relevant not only for mathematical modeling but also real weather scenarios, are involved here. To what extent must chaotic processes be negligible and when do these cases appear?

2) The GLM is studied and used for the main argument of the paper. But to what extent is the model relevant for the solution of typical sets of Navier-Stokes equations used in NWP? That is – the author should show how long time scale prediction of, for example, TC:s using typical NWP models can be motivated from the GLM. 

Detailed comments:

1)  A first person narrative is used. Is questionable here.

2) The paper is overflowed with abbreviations that are not but should be defined/explained beforehand like AEW, AEJ, 3DLM etc. 

3) Line 34: provide reference to the claim about the two week predictability limit.

4) Which basic numerical models have been used in this paper? The mere mentioning of EMD and EEMD is insufficient for the general reader.

5) The description of the GLM is a bit sloppy. Apart from the X, Y, Z variables, what do Xivariables etc signify?

6) Lines 310-311: Why don’t chaotic processes dominate? What is required for non-chaotic processes to assure long-term predictability?

7) Bottom of p. 11: this is hardly a typical example of a stiff problem.

8) What is here meant by “downscaling”? Define, describe!

Author Response

Dear Reviewer:

Attached please find responses in a PDF file.

Thanks very much,

-Bowen

Reviewer 2 Report

Review of 

On the Predictability of 30-day Global Mesoscale Simulations of 

African Easterly Waves in 2006: A View with the Generalized Lorenz Model

Authors: Bo-Wen Shen *

The article describes a research area which is of high interest since one likes to extend the weather forecast range from a period of 14 days to 30 days or more. The author introduces new analysis methods and also explains the theoretical background. 

In addition, he got new results concerning the role of AEW for formation of tropical cyclones. 

Thus, I recommend the publication of this article in the journal Geosciences. 

Minor remarks: 

Abstract line 10  please explain the acronyms AEW and AEJ

Figure 1 :  time scale unit of Figure 1b is unclear

            color table for Figure 1d is missing

line 109:  it is unclear how AEJ shows up in Figures 1c,d)

line 130:  please add a point before „Thus … 

line 176   did you explain the acronym IC before? 

line 269   please give a reference or an explanation for „Rayleigh parameter’’

line 385  I cannot see a black circle in Figure 1c)

Thank you for consideration!

Author Response

(The authors gave the same response as above.)
